# How Supportive Ethical Relationships Are Negatively Related to Palliative Care Professionals’ Negative Affectivity and Moral Distress: A Portuguese Sample

**DOI:** 10.3390/ijerph19073863

**Published:** 2022-03-24

**Authors:** Francisca Rego, Valentina Sommovigo, Ilaria Setti, Anna Giardini, Elsa Alves, Julliana Morgado, Marina Maffoni

**Affiliations:** 1Faculty of Medicine, University of Porto, 4200-319 Porto, Portugal; mfrego@med.up.pt (F.R.); elsamma.alves@gmail.com (E.A.); 2Department of Brain and Behavioural Sciences, Unit of Applied Psychology, University of Pavia, 27100 Pavia, Italy; valentina.sommovigo@unipv.it (V.S.); ilaria.setti@unipv.it (I.S.); 3Department of Management, University of Bologna-Rimini Campus, 47900 Rimini, Italy; 4IT Department, Istituti Clinici Scientifici Maugeri IRCCS, 27100 Pavia, Italy; anna.giardini@icsmaugeri.it; 5Institute of Philosophy and Human Sciences, Federal University of Pará, Belém 66075-110, Brazil; jullianamorgado@hotmail.com; 6Psychology Unit of Montescano Institute, Istituti Clinici Scientifici Maugeri IRCCS, 27040 Montescano, Italy

**Keywords:** moral distress, ethical climate, negative affectivity, healthcare professionals

## Abstract

In the modern healthcare landscape, moral distress has become an increasingly common phenomenon among healthcare professionals. This condition is particularly prevalent among palliative care professionals who are confronted with bioethical issues in their daily practice. Although some studies described the effects of poor ethical climate and negative affectivity on moral distress, how these variables could be incorporated into a single model is still unclear. Thus, this study aims to investigate whether ethical relationships with the hospital could be related to the intensity and frequency of moral distress, both directly and as mediated by professionals’ negative affectivity. Sixty-one Portuguese palliative care professionals completed web-based self-report questionnaires. After exploring descriptive statistics, mediation analyses were performed using the partial least squares method. The results indicated that the presence of positive relationships with the hospital reduced the professionals’ negative affectivity levels. This, in turn, led palliative care professionals to experience a lower frequency and intensity of moral distress. Being a physician was positively associated with negative affectivity but not with the frequency of moral distress. Considering the protective role of ethical relationships with hospitals, health organizations could consider implementing interventions to improve hospitals’ ethical climate and provide staff with ethics training programs.

## 1. Introduction

Palliative care aims to improve the quality of life of seriously ill patients and their families through the relief of pain and physical, psychological, social, and spiritual suffering. It is based on a personalized, holistic, and transcultural approach, focusing on patient-centered decision-making, communication, continuity of care, symptom management, and support for family members and healthcare professionals [1,2,3,4,5]. The healthcare professionals working in palliative care frequently face end-of-life and/or death situations and ethical dilemmas in their daily clinical practice. As a result, palliative care professionals are frequently exposed to psychological, emotional, and existential suffering, which puts them at particular risk of experiencing moral distress [4,6,7,8,9,10,11]. 

Andrew Jameton (1984) was the first to describe moral distress as a negative experience affecting nurses who, despite knowing the morally correct action to take, cannot act according to their morality, beliefs, and values due to different external constraints [7,9,12,13]. Moral distress encompasses constraints on healthcare professionals’ moral identities, responsibilities, and interpersonal relationships [14]. Thus, moral agency is affected by individual beliefs and desires and by external principles and rules related to social, cultural, and relational contexts [14,15,16,17]. Moral distress is accompanied by the impairment of the professional’s moral integrity and a sense of failure at his or her inability to act according to his or her ethical standards [7,13,18]. This phenomenon may be related to dissatisfaction with one’s work, compromising patient care [4,19]. To date, most previous studies have mainly measured moral distress using an overall index of distress. However, moral distress can also be assessed in terms of intensity and frequency [20,21,22]. Indeed, an ethical issue might be morally distressing both because it occurs many times and/or because it is perceived as having a serious impact on the professional’s well-being. The focus on these two dimensions may be particularly crucial in developing tailored interventions. However, most previous studies have examined these dimensions (i.e., intensity and frequency) together as an overall indicator of moral distress without differentiating between them. This study aims to bridge this gap.

Besides the intensity and frequency of this kind of malaise, it is also crucial to focus on what may ignite it. Indeed, moral distress among palliative care professionals may be attributed to manifold aspects: relationships with patients and caregivers, including the presence of a conspiracy of silence, communication problems, the desire for unnecessary aggressive treatments or palliative sedation and a lack of knowledge of patient values and desires, which cause pain to the patient; factors related to the organizational environment, including the presence of a poor ethical climate, hospital policies, inadequate staffing, and pressures to decrease costs; legal/ethical concerns, including fear of being judged by colleagues, and a lack of adequate knowledge of palliative care and bioethics; and personal issues, including the management of negative emotions and complexity in decision-making for the implementation of appropriate palliative care treatments [4,7,23,24,25,26]. Consequently, palliative care professionals frequently experience situations of powerlessness and interpersonal conflict in dealing with end-of-life issues and work-related constraints, such as problems in disclosing information to patients and families, restraints imposed by excessive workload, and ethical conflicts [4,7,19,27].

Thus, moral distress may be induced significantly by end-of-life-care issues and is associated with both internal (individual) and external (environmental) factors [3,28,29,30,31]. With regards to internal facets, the experience of negative emotions is common, especially for those caring for dying patients, and these emotions, such as powerlessness, frustration, guilt, loss, and grief, may be associated with moral distress perceptions [7,18,23,32,33,34]. When professionals are confronted with ethical dilemmas, feelings of despair and incapacity may emerge if they are incapable of making ethical decisions when providing care. In this vein, the evaluation and monitoring of healthcare professionals’ negative affectivity, which is the tendency to experience negative emotions over time and in different situations, may also be crucial for managing moral distress.

Concerning environmental factors, moral distress may also arise when an organization’s environment poses an additional burden on healthcare professionals, namely when the hospital climate is characterized by poorly supportive relationships with colleagues and/or superiors, moral dilemmas, and pervasive experiences of unethical behaviors [3,7,11]. Specifically, the hospital ethical climate can be defined as health professionals’ perceptions of the ethics-related atmosphere, organizational practices, procedures, and policies that may have ethical content and moral outcomes due to the potential impact on ethics-related actions involved in patient care and interpersonal relationships [3,35]. This is related to the implementation of ethical norms, which help build a work environment in which professionals follow ethical principles that support the quality of the health services they provide. In such workplaces, professionals practice ethical values that promote feelings of ownership and decrease feelings of loneliness, resulting in increased productivity and patient satisfaction with care [36,37,38]. Previous literature showed that numerous organizational and relational variables that contribute to creating a positive ethical climate can play a crucial role in protecting against moral distress. Among these, a sense of security and shared decision-making based on recognized ethical values, as well as openness of communication, ensure that individuals are supported to make decisions that are right for them. Additionally, other factors, including the maximization of the professional’s (potential and continuous) learning, cooperative and respectful teamwork based on humanized care, and an inclusive work environment, shape a positive ethical climate, whose presence positively influences the success of treatment as well as healthcare professionals’ performance and satisfaction with their job [39,40,41]. More specifically, given that palliative care professionals work in interdisciplinary teams, high-quality communication among team members and the development of individualized care plans to coordinate care contribute to creating a good hospital ethical climate, reducing healthcare professionals’ moral distress [3,35]. In this study, we focused on a specific facet of ethical climate, namely ethical relationships with the hospital. These refer to the extent to which the professional believes that hospital mission and policies help him or her face difficult patient care issues and offer a setting wherein everyone’s feelings are considered, conflicts are addressed openly, and clinical practice can be performed as he or she believes it should be [35]. Throughout this paper, the term ethical relationships will refer to this specific component of ethical climate which, by definition, we considered to be an essential factor potentially influencing moral distress.

Thus, in light of current knowledge, it is crucial to better understand how environmental and subjective variables may combine to exacerbate or prevent each other’s effects in terms of moral distress. In this regard, the literature and clinical practice depict ethical relationships with the hospital and negative affectivity as significant facets. Currently, although some studies have described the effects of poor ethical climate and negative affectivity on moral distress [10,28,42], how these variables might be incorporated into a single model is still unclear. Therefore, based on the above-discussed literature, we aimed to investigate whether ethical relationships with the hospital (i.e., independent variable) are related to the intensity and frequency of moral distress (i.e., dependent variables). Using a cross-sectional design, we also examined which role negative affectivity plays in this relationship. More specifically, the current research aimed to answer the following questions: Are the ethical relationships with the hospital directly related to the frequency of moral distress? Are the ethical relationships with hospitals directly related to the intensity of moral distress? Moreover, do these relationships also occur indirectly, as mediated by negative affectivity, while controlling for occupation?

## 2. Materials and Methods

### 2.1. Participants and Procedures

The questionnaires were administrated through a web-based approach (i.e., emails) to healthcare professionals working in two Portuguese hospitals and the members of the Portuguese Association of Palliative Care, This latter includes most of the healthcare professionals working in the field of palliative care at national level. Data were collected between February and April 2020. After providing their informed consent online, participants who voluntarily agreed to participate took approximately 15 minutes to complete the survey. To participate, respondents were required to be healthcare professionals working in Portuguese hospitals, be employed in palliative care in direct contact with patients and provide an informed consent form. The contact details of the researchers were available in case any questions or clarifications were needed. The participation of healthcare professionals was voluntary and anonymous.

### 2.2. Ethical Considerations

This research is part of a larger European research project called WeDistress HELL (wellness and distress in healthcare professionals dealing with end-of-life and bioethical issues), which was reviewed and approved by the Ethics Committee of ICS Maugeri—Institute of Pavia, Italy (Protocol No. 2211CE, 19 June 2018). All participants provided their informed consent before participating in the study. The data in this study were collected and analyzed anonymously.

This study was also submitted to and approved by the Ethics Committees of the Hospital Center Trás-os-Montes e Alto Douro and Hospital Center Tâmega e Sousa.

### 2.3. Measures

The current research is based on well-established instruments that have been broadly utilized in the literature to evaluate the study’s constructs.

Ethical relationships within the hospitals were assessed using the six-item subscale of the Hospital Ethical Climate Survey (HECS; Olson, 1998). The responses were rated on a five-point Likert scale (1 = almost never true, 5 = almost always true), where higher scores indicate a more positive perception of ethical relationships with the hospital. This measure has been translated into different languages consistently showing good internal consistency and construct validity [43,44,45]. 

Negative affectivity (NA) was evaluated using the ten-item subscale of the Positive and Negative Affect Schedule(PANAS) [46] in its Portuguese version [47]. Professionals were asked to indicate how frequently they generally experienced each of the ten negative emotional states on a five-point Likert scale (0 = very slightly or not at all to 5 = extremely), where higher scores indicate higher levels of negative affectivity. PANAS has been widely used in different contexts, including the healthcare sector [10,48], showing excellent psychometric properties.

Moral distress was measured using the *Moral Distress Scale-Revised* [49]. Its Portuguese version [50] includes two subscales: (a) frequency, which refers to how frequently professionals experienced situations that might generate moral distress, such as having to provide inappropriate treatments to their patients or to behave in a questionable ethical way during their daily clinical practice; (b) intensity, which refers to the degree to which the experienced situation was perceived as disturbing. Each of the twenty-one items is scored on a five-point Likert scale in terms of frequency (0 = never, 4 = very frequently) and intensity (0 = none, 4 = great extent). The scores for each subscale were calculated by summing their items, obtaining scores ranging from 0 to 84. This scale has been used extensively in prior investigations on healthcare providers employed in intensive care units [21,51], showing good internal consistency [21].

### 2.4. Instruments Translation

Dr. Olson, the original author of the HECS, was contacted and authorization was granted to develop a Portuguese version of this instrument. The translation of the HECS was performed according to the standard guidelines for translating questionnaires [52]. Two independent bilingual translators translated the items into their mother tongue—Portuguese. The forward translation was then reviewed by a bilingual expert panel that selected appropriate options for inappropriate expressions. An independent translator, who was not involved in the initial translation, proceeded with the back-translation. Both Portuguese- and English-speaking natives made a comparison between the back-translated version and the original version of the scale, and further changes were implemented. Finally, a preliminary test was conducted. Even though the original scale contained wording specifically targeted at hospital nurses only (i.e., “nurses”), we decided to utilize the more general term “healthcare professionals” to make it suitable for all providers working within the hospital setting. This revised form received the approval of the author of the original scale and was also utilized in its Italian validation [53]. 

### 2.5. Statistical Analysis

Firstly, the data were explored for descriptive statistics and intercorrelations among the study’s variables using SPSS IBM SPSS statistics 23 [54]. Next, independent sample t-test analyses and analyses of variance (ANOVAs) were performed to detect differences in the study’s variables between groups that differed in gender, occupation, educational level (i.e., Bachelor’s degree vs. Master’s degree), type of contract (open-ended vs. fixed-term contract), type of work (shift work vs. non-shift work), exposure to extra-organizational stressful events, age, years of overall experience and years of experience in the current position, considering Cohen’s d values (d = 0.2 represents a small effect size, d = 0.5 a medium effect size and d = 0.8 a large effect size [55]). Next, to test whether ethical relationships within the hospital were associated with both the frequency and intensity of moral distress through negative affectivity, we decided to adopt partial least squares (PLS) structural equation modeling (SEM). This method was considered particularly suitable for our sample size. Indeed, it is a variance-based SEM that, even using small sample sizes, enables us to reach higher statical power levels and shows much better convergence behavior compared to covariance-based (CB-) SEM [56,57]. It calculates the relations between all variables simultaneously and does not require multivariate normality [58]. Thus, using SmartPLS v. 3.2.6. [59], we applied the repeated indicator approach in a reflective-formative type to test the measurement model. Loading values were considered acceptable when equal to or higher than 0.70 [60]. The composite reliability of the study’s constructs was considered good when equal to or greater than 0.80 [61]. The convergence validity was evaluated based on the average variance extracted (AVE) values, such that it was deemed good when the AVE values were lower than 0.50 [62]. The Fornell-Larcker discriminant validity criterion (i.e., for any latent variable, the square root of AVE should be higher than its correlation with any other latent variable, ([63], p. 67)) was used to evaluate the discriminant validity among the study’s variables. The structural model was computed using a bootstrapping procedure (500 subsamples). Furthermore, as indexes of model consistency and predictive relevance, we considered the R^2^ (i.e., coefficient of determination; 0.75, 0.50, 0.25, respectively, refer to substantial, moderate, or weak levels of predictive accuracy [64] and Q^2^ (i.e., cross-validated redundancy; values larger than zero indicate significant predictive relevance; [64]) values. 

## 3. Results

Sixty-one healthcare professionals participated in this study. Most of the respondents were women (85.2%) with an open-ended contract (83.2%) and a Master’s degree (59.0%) who worked day shifts (65.6%; see Table 1). Most of the respondents were over 41 years of age (54.0%), with an overall job tenure between 6 and 15 years (34.4%) and a job tenure in their current position greater than 10 years (37.7%). The participants were employed primarily as physicians (42.6%), followed by nurses (34.4%), psychologists (11.5%), social workers (6.6%), social-health workers (3.3%), and physiotherapists (1.6%). Approximately 64% of the respondents stated that they had been exposed to stressful extra-work events in the previous year, including family problems (23%), mourning (8.2%), moving (4.9%), illnesses (1.6%), work-related problems (1.6%), and outbreak-related concerns (1.6%). 

The descriptive statistics and correlations among the study’s variables are shown in Table 2. Professionals’ negative affectivity statistically significantly correlated with ethical relationships with the hospital (r = −0.46, *p* < 0.001), while it was negatively related to both the frequency (r = 0.36, *p* < 0.01) and the intensity (r = 0.22, *p* < 0.05) of moral distress. These latter two dimensions were statistically significantly and positively associated with each other (r = 0.41, *p* < 0.05). Contrary to our expectations, ethical relationships within the hospital were statistically significantly related to neither frequency (r = −0.23, ns) nor intensity (r = −0.15, ns) of moral distress. In contrast to our expectations, being a physician was the only demographic variable that was statistically significantly associated with some of the study’s variables. More specifically, being a physician (rather than another professional) was positively associated with both negative affectivity (r = 0.37, *p* < 0.01) and the frequency of moral distress (r = 0.27, *p* < 0.05). As a result, according to recommended practices [65] and previous studies [66], we decided to use only those that significantly correlated with the variables of interest as control variables in our subsequent statistical analyses. Accordingly, we controlled negative affectivity and the frequency of moral distress for the occupation of the physician (other occupations = 1, occupation of physician = 2). 

The results of the independent *t*-test analyses (see Table 3) indicated that there were no statistically significant differences with regard to gender, type of contract (i.e., open-ended vs. fixed-term contract), shift work (i.e., workers vs. shift workers), exposure to stressful extra-organizational events (i.e., non-affected vs. affected), or educational level (i.e., Bachelor’s degree vs. Master’s degree). Conversely, significant differences were found for occupation: the physicians reported that they perceived greater negative affectivity (M = 2.07, SD = 0.72) and experienced moral distress more frequently (M = 28.50, SD = 15.84) than other professionals (negative affectivity: M = 1.58, SD = 0.52; frequency of moral distress: M = 21.14, SD = 11.29; see Figure 1 and Figure 2). The Cohen’s d values indicated, respectively, large (negative affectivity: d = 0.78) and medium (frequency of moral distress: d = 0.55) effect sizes for these differences. The results of the ANOVAs showed that there were no statistically significant differences between the groups with different ages (negative affectivity: F(3,58) = 1.70, ns; ethical relationships: F(3,58) = 0.44, ns; frequency of moral distress: F(3,58) = 0.54, ns; intensity of moral distress: F(3,58) = 0.64, ns); years of overall experience (negative affectivity: F(3,58) = 1.73, ns; ethical relationship: F(3,58) = 1.12, ns; frequency of moral distress: F(3,58) = 1.45, ns; intensity of moral distress: F(3,58) = 2.25, ns) or years of experience in the current position (negative affectivity: F(3,58) = 0.87, ns; ethical relationships: F(3,58) = 0.26, ns; frequency of moral distress: F(3,58) = 0.76, ns; intensity of moral distress: F(3,58) = 0.92, ns).

Following previous authors [67], a total score for negative affectivity traits (scores ranging from 10 to 50) was calculated and two groups (low vs. high negative affectivity) were created based on the average level of negative affectivity traits reported in the Italian validation of PANAS (i.e., 20.90 [68]). Healthcare professionals with different (low vs. high) negative affectivity levels differed in their perceptions of ethical relationships with their hospital and the frequency and intensity of moral distress (as depicted in Figure 3 and Figure 4), such that healthcare professionals with high negative affectivity levels reported lower perceptions of ethical relationships with their hospital and higher levels of frequency and intensity of moral distress than those with low negative affectivity. 

To check the appropriateness of our sample size, we performed a power analysis for a multiple regression analysis with three predictors (i.e., ethical relationships with the hospital, negative affectivity, occupational group) with the program G*Power. The results of this analysis, which was performed using an alpha of 0.05, a power of 0.95, and a medium effect size, indicated that a sample of at least 49 subjects was required, suggesting that our sample size was adequate. Next, using SmartPLS v. 3.2.6. [59], we tested the measurement model which, based on our hypotheses, included two reflective constructs (i.e., ethical relationships within the hospital, negative affectivity) and two sub-scores of moral distress (i.e., frequency and intensity) resulting from the total sum that was obtained from each item in their respective 21 questions. All the items had statistically significant and acceptable loading values (>0.70; [60]). The composite reliabilities of the study’s constructs were satisfactory, since the values were greater than 0.80 and lower than 0.95 [69]. The convergence validity was good because all the average variance extracted (AVE) values were above the suggested value of 0.50 [62]. The fact that the correlations between each pair of latent constructs did not exceed the square root of each construct’s AVE provided further support for the discriminant validity of our constructs [70]. Next, the structural model was computed. The structural coefficients presented in the PLS model (see Table 4 and Figure 5) that ethical relationships within the hospital were statistically significantly and negatively related to negative affectivity (β = −0.43, t = 3.61, *p* < 0.001, 95 CI [−0.61, −0.24]). Negative affectivity was statistically significantly and positively associated with both the frequency (β = 0.33, t = 2.45, *p* < 0.01, (0.13, 0.54)) and the intensity (β = 0.24, t = 1.91, *p* < 0.05, (0.03, 0.44)) of moral distress. The occupation of the physician was positively and statistically significantly associated with negative affectivity (β = 0.29, t = 2.48, *p* < 0.01, (0.10, 0.48)), but not with the frequency of moral distress (β = 0.14, t = 1.05, ns, (−0.10, 0.32)). The results of mediation models indicated that negative affectivity mediated the associations of ethical relationships within the hospital with both the frequency (β = −0.14, t = 2.08, *p* < 0.05, (−0.25, −0.06) and intensity (β = −0.10, t = 1.70, *p* < 0.05, (−0.20, −0.02)) of moral distress. The presence of ethical relationships within the hospital setting reduced professional negative affectivity levels. This, in turn, led professionals to experience a lower frequency and intensity of moral distress. 

The indicators of consistency were appropriate, even though a small amount of variation in the constructs of interest was found (R2 (negative affectivity) = 0.32; R^2^(frequency of moral distress) = 0.16; R^2^(intensity of moral distress = 0.06). The predictive relevance of the indicators (Q2 (negative affectivity) = 0.14; Q^2^ (frequency of moral distress) = 0.12; Q^2^ (intensity of moral distress) = 0.04) were greater than zero, indicating that the model was relevant to the prediction of these constructs [64].

Note. 95% CI = confidence intervals; Ethical relationships = ethical relationships within the hospitals; NA = negative affectivity, Frequency_MDS = frequency of moral distress; Intensity_MDS = intensity of moral distress; role = other occupations vs. occupation of physician.

## 4. Discussion

Moral distress can be considered an increasing challenge for the healthcare system, especially for palliative care professionals. Indeed, these professionals are frequently exposed to manifold moral quandaries in their daily clinical practice. Although previous literature pinpointed ethical relationships with hospital and negative affectivity as facets that can impact this kind of malaise [3,7,11,33], the question of how these variables may combine to explain the development of moral distress is understudied. In this regard, the current study found that ethical relationships with the hospital were negatively related to Portuguese palliative care professionals’ negative affectivity levels. This, in turn, led professionals to experience a lower frequency and intensity of moral distress. Hence, several findings deserve to be discussed considering the current knowledge. 

Firstly, ethical relationships with the hospital can be assumed to be a feature of the ethical climate that may reduce palliative care professionals’ negative affectivity levels. This finding is consistent with previous studies reporting the importance of adopting an environmental perspective to better understand the quality of professionals’ affectivity and work experiences [10,67,71,72]. Specifically, in her theoretical paper, Corley [72] suggested that a positive ethical climate where decisions are shared and responsibilities are distributed among colleagues may help professionals to make correct decisions and perform correct actions, positively impacting their affectivity. Moreover, the literature identified the presence of an ethical environment and respectful relationships as key factors promoting virtuous organizational dynamics, which may result in higher levels of well-being and positive affectivity among professionals [73,74]. This may be because ethical relationships within the workplace may help professionals not only to make shared decisions and provide respectful care but also to accept and face challenging situations, reducing their consequent distress and negative emotional responses [10,42,67]. 

Secondly, this study deepens our understanding of the mechanisms explaining the effects of ethical relationships on moral distress. Indeed, our results showed that the beneficial effects of ethical relationships were not transmitted to moral distress directly but, rather, mediated by professionals’ self-reported negative affectivity. Thus, ethical relationships were negatively related to professionals’ negative affectivity. This, in turn, was negatively associated with the frequency and intensity of moral distress. Although further studies are necessary to replicate this finding, it is possible to speculate that negative affectivity may work as a sort of internal catalyst igniting the experience of moral distress. The distress of individual professionals could be alleviated when this emotional trigger is reduced thanks to positive ethical relationships with the hospital. Moreover, unlike most previous studies, which used a general index of malaise to assess moral distress, we considered both the frequency of occurrence and the intensity of morally distressing situations experienced by palliative care professionals. In doing so, we provided a more fine-grained understanding of these two components of moral distress. Additionally, by identifying, for the first time, negative affectivity as a variable that helps explain the association between ethical relationships with the hospital and moral distress, we contributed to an increasing body of research analyzing the link between the two constructs [10,42,67,75,76,77,78]. 

Finally, compared to other professionals, physicians were more likely to report higher negative affectivity levels and experience morally distressing situations more frequently. Although there have been mixed results on health occupations at higher risk of moral distress, most studies found higher levels of moral distress in nurses than in physicians; this was probably due to the considerable amount of time that nurses generally spend in direct contact with patients [28,49]. A plausible explanation for our unexpected results was that, given our limited sample size, we compared the self-reported experiences of physicians with those of different healthcare professionals. Furthermore, our group comparisons, which were based on socio-demographic variables, did not indicate any statistically significant differences across gender, educational levels, or employment statuses. This was in line with the results of previous studies [67,76] and might also be linked to our small and probably not sufficiently balanced sample. Similarly, in the current sample, no differences were reported concerning shift work or exposure to stressful extra-organizational events. These data diverge from a previous Italian study, in which palliative care and neuro-rehabilitation professionals performing shift work or experiencing stressful extra-work events reported lower ethical climate perceptions [67]. These nonsignificant results might be explained by considering that a work environment characterized by the existence of a respectful approach to patient care and ethical relationships between healthcare professionals involved in the provision of patient care may shape a breeding ground for all professionals regardless of their socio-demographic differences [10]. Similarly, caring for a patient with life-threatening diseases may potentially be experienced as a morally distressing situation regardless of the professional’s personal characteristics and hours spent in the unit. Conversely, the specific profession seems to be what can make the difference. In this view, physicians perform different tasks to other healthcare professionals. For example, they are asked to make decisions and assume responsibility for actions related to patient treatment [67], which can contribute to explaining why physicians reported experiencing morally distressing situations more frequently.

However, the results of this study should be considered in light of its limitations. 

First, because of the cross-sectional nature of this study, no causal inference can be derived. For instance, it is still unclear whether palliative care professionals can feel ashamed or nervous because they cannot act morally or rather because of organizational practices, procedures, and policies in their hospital. Second, this study relies on self-reported measures and suffers from the limitations of this methodology (e.g., social desirability bias). Thus, future studies should adopt a longitudinal design and integrate information from different sources using diverse data collection methods. To this end, future studies should integrate quantitative data with qualitative data obtained through focus groups and in-depth interviews with Portuguese healthcare professionals working in palliative care. This will make it possible to reach a better understanding of the state of moral distress experienced by this specific population. Third, selection bias cannot be ruled out due to the voluntary participation of the respondents in this study. For instance, given the “healthy worker effect” [79], it might be that the healthcare professionals who took part in the research were healthy enough to remain in the jobs, while those who were greatly affected by moral distress might have been absent from work due to their poor health condition. Future research should include an incentive for respondents to encourage broader participation to limit this bias. Fourth, this study was limited to a very small sample from the Portuguese palliative care context, so replications in larger, more nationally representative samples would help increase the generalizability of our findings. Additionally, future research should focus on different subgroups of healthcare professionals to deepen the reasons why some occupational groups (e.g., physicians) might be the most at risk of moral distress. Moreover, since previous studies indicated the presence of cross-national differences in the development of psychological outcomes [80], replications should also be conducted in other cultural contexts. 

Fifth, previous studies revealed the presence of considerable differences across different hospitals, identifying hospital-level perceptions of workplace climate as the strongest antecedent of health outcomes [81]. Thus, healthcare professionals working in the same hospital would be more similar with regards to their perceptions of ethical climate and moral distress [53]. Although our research participants were from different hospitals, we could not conduct multi-level analyses at the hospital level because of our small sample size. A multi-level approach is recommended for future studies to appropriately analyze whether the hospital-level ethical climate might contribute to a certain ethical climate within particular teams and might, therefore, influence individual outcomes.

Sixth, since moral distress has been previously described as multifactorial in causation, future studies are recommended to explore some known factors, such as communication problems, lack of resources, witnessing professionals giving false hope to patients and factors related to family members [4].

Last, but not least, these data were collected before and, partially, during the fisrt phase of the COVID-19 pandemic.. As a result of the outbreak, healthcare workers were confronted with ethical issues that had not been previously experienced (e.g., patient prioritization), which might have further increased their vulnerability to moral distress. Thus, replications are needed during and after pandemic times.

Despite these limitations, the findings of this study reveal the importance of preserving and fostering positive ethical relationships because of their beneficial role in preventing professionals from experiencing negative affectivity and moral distress. Thus, from a practical standpoint, human resource managers should implement practices to improve the ethical climate, such as by introducing a code of ethics and standardized procedures that encourage reflection about ethical concerns and rewarding interdisciplinary communication about patient care quality [10,82]. Managers could also consider introducing “ethics rounds” to support healthcare professionals in handling ethically difficult situations [10,83]. To this end, ethical dialogue with senior professionals and reflection on ethically critical aspects of patient cases within inter-professional expert groups should be encouraged [10]. Moreover, health organizations should provide palliative care professionals with ethics training programs to educate them on how to face daily ethical issues by fostering their ethical and relationships skills [83]. These programs should be integrated with specific training aimed at strengthening positive affectivity, such as brief present-moment awareness [84]. Specific training programs for supervisors on how to effectively support their staff in the face of critical events and how to deal with errors constructively could also be useful for further reducing negative affectivity [10].

## 5. Conclusions

This study enriches the literature by clarifying through which mechanism the ethical relationships with hospitals were negatively related to moral distress in a sample of Portuguese palliative care professionals. The results indicated that ethical relationships within the hospitals were negatively related to the professionals’ negative affectivity levels which, in turn, decreased the frequency and intensity of moral distress. Thus, organizational interventions and ethics training programs aimed at fostering a positive hospital ethical climate and reducing professionals’ negative affectivity must be considered an urgent imperative to preserve the well-being of healthcare professionals at present and in the future. 

## Figures and Tables

**Figure 1 ijerph-19-03863-f001:**
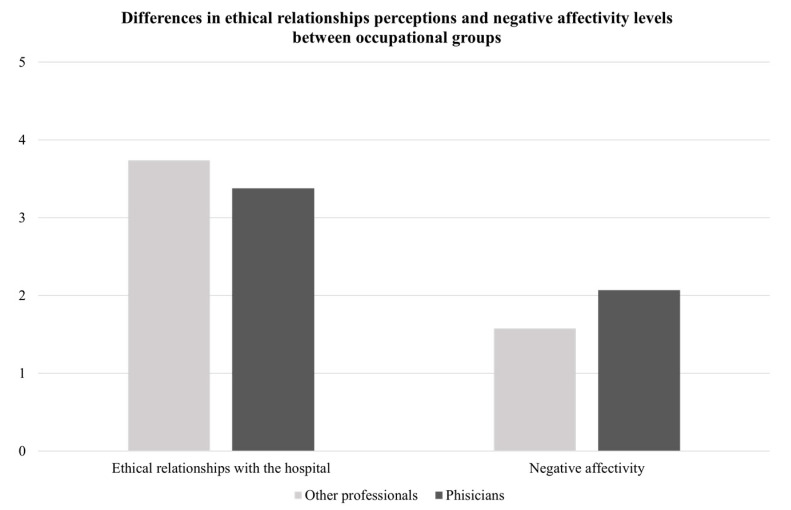
Differences in ethical relationships with the hospital and negative affectivity between occupational groups.

**Figure 2 ijerph-19-03863-f002:**
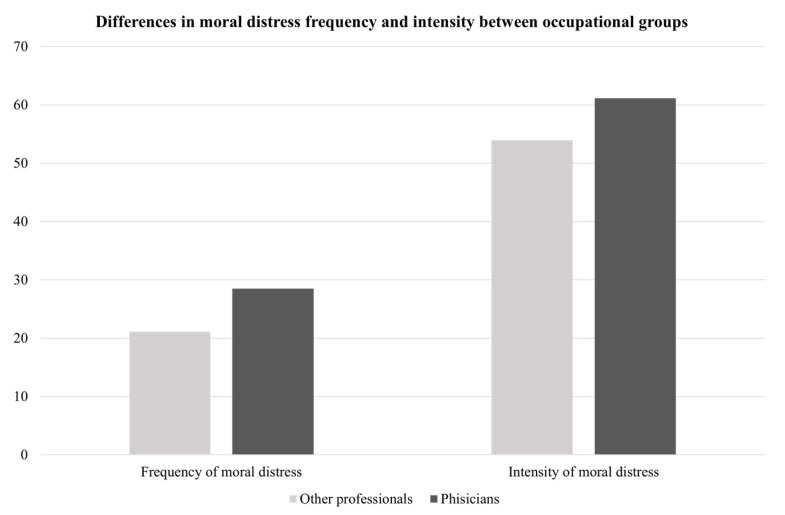
Differences in moral distress frequency and intensity between occupational groups.

**Figure 3 ijerph-19-03863-f003:**
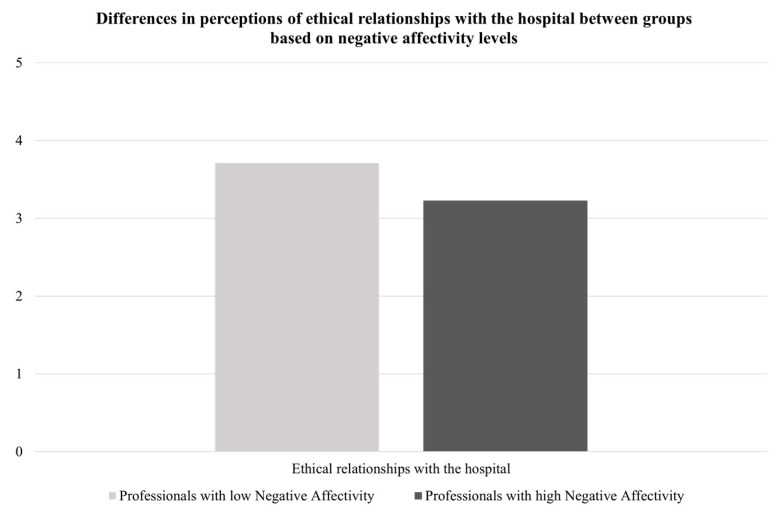
Differences in perceptions of ethical relationships with the hospital between groups based on negative affectivity levels.

**Figure 4 ijerph-19-03863-f004:**
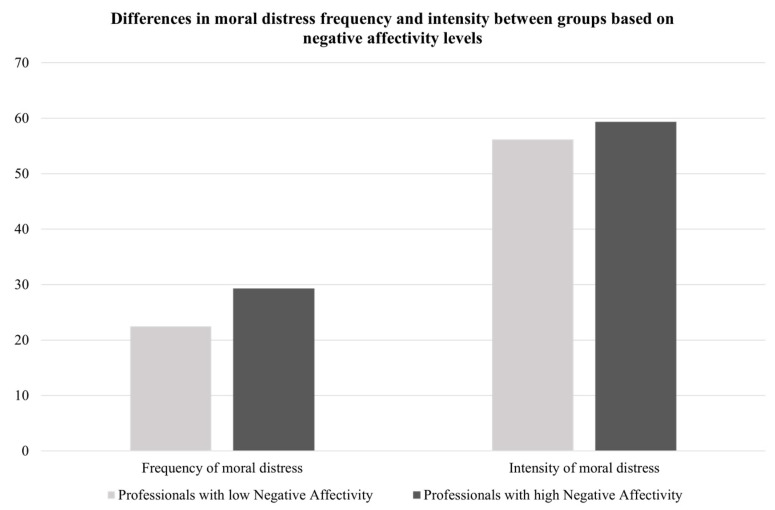
Differences in perceptions of moral distress based on negative affectivity levels.

**Figure 5 ijerph-19-03863-f005:**
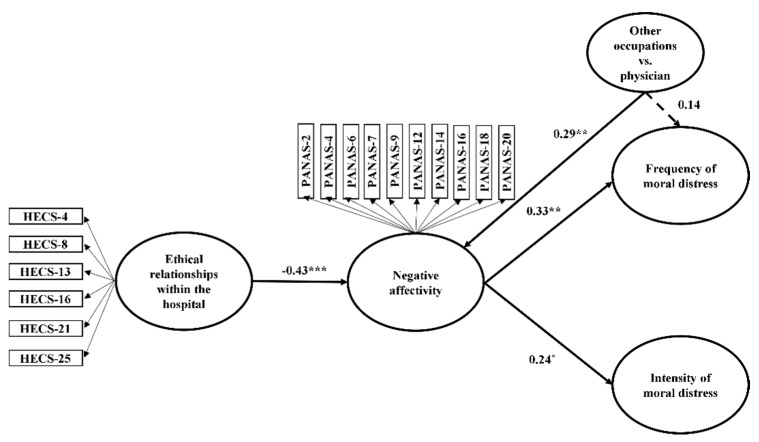
The mediation model with negative affectivity as a mediator of the associations between ethical relationships within the hospitals and both frequency and intensity of moral distress, while controlling for occupation. Note. * *p* < 0.05; ** *p* < 0.01; *** *p* < 0.001.

**Table 1 ijerph-19-03863-t001:** Descriptive statistics regarding the sample (*n* = 61).

Variable	*n*	%
**Gender (%)**		
Female/Male	52/9	85.2/14.8
**Age (%)**		
<30 years	4	6.6
30–40 years	24	39.3
41–50 years	16	26.2
>51 years	17	27.8
**Education (%)**		
Bachelor’s degree	22	36.1
Master’s degree	39	63.9
**Occupation (%)**		
Physician	26	42.6
Other professionals ^1^	35	57.4
**Overall job tenure (%)**		
6–15 years	21	34.4
16–25 years	17	27.9
26–30 years	6	9.8
>30 years	17	27.9
**Job tenure in current position (%)**		
<5 years	21	34.4
6–10 years	17	27.9
>10 years	23	37.7
**Shift work (*%*)**		
Yes-No	21-40	34.4-65.6
**Contract (*%*)**		
Open-ended	52	83.2
Fixed term	9	14.8
**Stressful event**		
Yes-No	39-22	63.9-36.1

^1^ Other professionals included psychologists (*n* = 7), nurses (*n* = 21), social workers (*n* = 4), social-health operators (*n* = 2), physiotherapists (*n* = 1).

**Table 2 ijerph-19-03863-t002:** Descriptions, internal consistency and intercorrelations of the study’s variables (*n* = 61).

Measure	M	SD	rho_A	Composite Reliability	AVE	1	2	3	4	5	6	7	8	9	10	11	12
1. NA	1.79	0.65	0.89	0.91	0.53	**0.88**											
2. Hospital	3.59	0.75	0.85	0.87	0.51	−0.46 ***	**0.82**										
3. MDS_freq	24.28	13.80	-	-	-	0.36 **	−0.23	**0.91**									
4. MDS_int	57.02	24.63	-	-	-	0.22 *	−0.15	0.41 *	**0.97**								
5. Gender	-	-	-	-	-	−0.09	0.22	−0.05	−0.08	-							
6. Age	-	-	-	-	-	−0.17	0.19	−0.12	−0.10	−0.14	-						
7. Tenure_t	-	-	-	-	-	−0.22	0.21	−0.01	0.16	−0.26 *	0.44 **	-					
8. Tenure_p	-	-	-	-	-	−0.07	0.20	−0.02	−0.05	−0.18	0.47 **	0.12	-				
9. Education	-	-	-	-	-	0.04	−0.12	−0.12	0.04	0.15	−0.30 *	−0.15	−0.40 **	-			
10. Physician						0.37 *	−0.23	0.27 *	0.15	0.20	−0.03	−0.19	0.01	0.03	-		
11. Shift	-	-	-	-	-	0.15	−0.02	0.15	0.07	0.09	−0.04	−0.30 *	0.13	−0.18	0.21	-	
12. Contract	-	-	-	-	-	−0.03	−0.06	−0.14	0.02	−0.04	−0.24	0.27 *	−0.18	0.23	−0.08	−0.20	-
13. Event	-	-	-	-	-	0.15	−0.06	0.07	0.10	−0.17	−0.21	−0.23	0.11	0.04	0.09	0.18	0.02

Note. Boldfaced numbers on the diagonal represent Cronbach’s alpha; AVE = average variance extracted; M = means; SD = standard deviation;* *p* < 0.05; ** *p* < 0.01, *** *p* < 0.001; NA = negative affectivity; Hospital = ethical relationships with the hospitals; MSD_freq = frequency of moral distress; MSD_int = intensity of moral distress; Gender: 0 = female, 1 = male; age:1 = <30 years old, 2 = 30–40 years old, 3 = 41–50 years old, 4 = 51–60 years old, 5 = >60 years old; Tenure_t = job tenure in total: 1 = <5 years, 2 = 6–15 years, 3 = 16–25 years, 4 = 26–30 years, 5 = >30 years; Tenure_*p* = tenure in the current position: 1 = <5 years; 2 = 6–10 years; 3 = >10 years; Education: 1 = high-school degree, 2 = bachelor degree, 3 = master degree, 4 = PhD; Physician = others (=1) vs. role of physician (=2); Shift = 0 = no, 1 = yes; Contract = 1 = open-ended contract, 2 = fixed-term contract; Event = exposure to stressful extra-work event, 0 = no, 1 = yes; Rho_A, composite reliability and AVE were calculated using SmartPLS v. 3.2.6., whereas correlations and Cronbach’s alphas were computed using SPSS.

**Table 3 ijerph-19-03863-t003:** Means, standard deviations, t-values of the study variables across gender, occupation, type of contract, shift vs. non-shift work, exposure vs. non-exposure to extra-work stressful events.

	**Females** (***n*****= 52**)		**Males** (***n*****= 9**)		**t**	* **p** *	**95% CI**		**Cohen’s** * **d** *
	**M**	**SD**	**M**	**SD**			**LL**	**UL**	
Negative affectivity	1.81	0.64	1.64	0.74	0.72	0.47	−0.30	0.64	-
Ethical relationships	3.51	0.75	3.98	0.66	−1.72	0.09	−0.10	0.07	-
Frequency_MDS	24.57	14.15	22.56	12.10	0.40	0.69	−8.02	12.06	-
Intensity_MDS	57.88	25.03	52.00	22.81	0.66	0.51	−11.99	23.76	-
	**Other professionals** **(*n* = 35)**		**Physicians (*n* = 26)**		**t**	* **p** *	**95% CI**		**Cohen’s *d***
	**M**	**SD**	**M**	**SD**			**LL**	**UL**	
Negative affectivity	1.58	0.52	2.07	0.72	−3.06	0.05	−0.80	−0.17	0.78
Ethical relationships	3.74	0.64	3.38	0.85	1.85	0.07	−0.03	0.74	-
Frequency_MDS	21.14	11.29	28.50	15.84	−2.12	0.04	−14.31	−0.41	0.53
Intensity_MDS	53.94	27.33	61.15	20.22	−1.13	0.26	−19.94	5.52	-
	**Open-ended contract** **(*n* = 52)**		**Fixed term contract** **(*n* = 9)**		**t**	* **p** *	**95% CI**		**Cohen’s *d***
	**M**	**SD**	**M**	**SD**			**LL**	**UL**	
Negative affectivity	1.80	0.70	1.74	0.30	0.22	0.82	−0.42	0.53	-
Ethical relationships	3.57	0.76	3.70	0.73	−0.50	0.62	−0.68	0.41	-
Frequency_MDS	25.10	14.18	19.56	10.74	1.11	0.27	−4.41	15.49	-
Intensity_MDS	56.85	24.25	58.00	27.13	−0.13	0.90	−19.09	16.78	-
	**Workers** **(*n* = 40)**		**Shift workers (*n* = 21)**		**t**	* **p** *	**95% CI**		**Cohen’s *d***
	**M**	**SD**	**M**	**SD**			**LL**	**UL**	
Negative affectivity	1.72	0.61	1.92	0.72	−1.16	0.25	−0.55	0.15	-
Ethical relationships	3.60	0.75	3.56	0.77	0.18	0.86	−0.37	0.45	-
Frequency_MDS	22.82	13.62	27.04	14.04	−1.14	0.26	−11.64	3.20	-
Intensity_MDS	55.72	25.92	59.47	22.36	−0.56	0.58	−17.11	9.60	-
	**No extra-work** **stressor** **(*n* = 22)**		**Extra-work stressor** **(*n* = 39)**		**t**	* **p** *	**95% CI**		**Cohen’s *d***
	**M**	**SD**	**M**	**SD**			**LL**	**UL**	
Negative affectivity	1.66	0.73	1.86	0.60	−1.78	0.24	−0.55	0.14	-
Ethical relationships	3.52	0.84	3.62	0.71	−0.50	0.62	−0.51	0.30	-
Frequency_MDS	23.04	14.37	24.97	13.61	−0.52	0.60	−9.33	5.48	-
Intensity_MDS	53.37	26.34	58.92	23.74	−0.80	0.42	−18.47	7.89	-
	**Bachelor’s (*n* = 35)**		**Master’s degree (*n* = 26)**		**t**	* **p** *	**95% CI**		**Cohen’s *d***
	**M**	**SD**	**M**	**SD**			**LL**	**UL**	
Negative affectivity	1.72	0.65	1.82	0.66	−0.60	0.55	−0.46	0.25	-
Ethical relationships	3.69	0.78	3.52	0.74	0.79	0.43	−0.24	0.56	-
Frequency_MDS	26.04	12.51	23.28	14.54	0.75	0.46	−4.62	10.15	-
Intensity_MDS	58.59	24.74	56.13	24.84	0.37	0.71	−10.77	15.70	-

**Table 4 ijerph-19-03863-t004:** Effect of hospital ethical climate on both frequency and intensity of moral distress through negative affectivity, while controlling for role.

Effects	Original Sample	T Statistics	*p* Values	95% CI
Ethical relationships→Negative affectivity	−0.43	3.61	0.000	[−0.61, −0.24]
Negative affectivity→Frequency of moral distress	0.33	2.48	0.007	[0.13, 0.54]
Negative affectivity→Intensity of moral distress	0.24	1.91	0.028	[0.03, 0.44]
Role→Negative affectivity	0.29	2.48	0.007	[0.10, 0.48]
Role→Frequency of moral distress	0.14	1.05	0.147	[−0.10, 0.32]
Ethical relationships→Negative affectivity→Frequency of moral distress	−0.14	2.08	0.011	[−0.25, −0.06]
Ethical relationships→Negative affectivity→Intensity of moral distress	−0.10	1.70	0.044	[−0.20, −0.02]

## Data Availability

Data are available from the authors upon reasonable request.

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
