# Peer review of "How Supportive Ethical Relationships Are Negatively Related to Palliative Care Professionals’ Negative Affectivity and Moral Distress: A Portuguese Sample"

_ijerph, 2022, doi:10.3390/ijerph19073863_

Round 1

Reviewer 1 Report

Review Report

  1. Inadequate Sample: The sampling type/the sampling method, inclusion and exclusion criteria in the sample, and the source population for the sample are not clear.
  2. The sample size N=61 is too small. The study is not adequately powered. This affects the reliability of a survey's results because it leads to a higher variability, which may lead to bias.
  3. The research design, the research question/research hypothesis, and the objectives of the study are not clear. The dependent/outcome variable and independent variables are not clearly identified.
  4. In light of the small sample size, the study had not incorporated focus group discussions or in-depth interviews with health care professionals that could have provided better insight into the state of moral distress in the study population.
  5. Inadequate and non-diversified data visualization and presentation in the Results section and the lack of clear narrative summary of the findings makes it difficult to distinguish significant factors from non-significant factors affecting moral distress.  
  6. Moral distress is multifactorial in causation and this study does not include some known factors such as communication problems, lack of resources, witnessing professionals giving false hope to patient and factors related family members.1
  7. In collusion, this study falls short of adding to what we already know about moral distress, perhaps because too small a slice of a larger study was taken. The study does not produce new knowledge or expand our understanding of moral distress.

Reference

  1. Corradi-Perini C, Beltrão JR, Ribeiro URVCO. Circumstances Related to Moral Distress in Palliative Care: An Integrative Review. Am J Hosp Palliat Care. 2021 Nov;38(11):1391-1397. doi: 10.1177/1049909120978826. Epub 2020 Dec 7. PMID: 33280390.

Reviewer 2 Report

Thank you for the opportunity to review “From ethical climate to moral distress through the mediation of negative affectivity in a sample of Portuguese palliative care professionals.” The authors examine the relationship between ethical environment and moral distress (intensity and frequency of such concerns), which is an important aspect of oncology/palliative care. With a cross sectional study design, the authors use mediation analyses to help understand the relationship between variables of interest. Understanding this relationship will help develop interventions and increase opportunities for education (some other practical suggestions offered by the authors). Although sample size is small with majority of them being female, day shift personnel and non-physicians, the study offers insight into the interaction of variables that may contribute to moral distress. Larger studies to replicate findings and focus on different subgroups of healthcare professionals may be useful. Interestingly, the timing of study coincides with the time the COVID 19 pandemic originated. Given the study findings help shape strategies to address the problem of moral distress, I believe it is worthy of publication.

Reviewer 3 Report

From Ethical Climate to Moral Distress     Reviewer comments

This article looks at the correlations between 3 variables:

1) Moral Distress (frustration at the inability to act morally)

2) PANAS negativity (shame, fear, nervousness, etc.)

3) Hospital ethics (policies, mission, trust, communication, social support.)

Statistically, these 3 have only correlations, not causal connections, showing either that:

(A)  (3) Social support reduces (2) shame and fear in (1) violating one’s moral conscience  OR

(B)  (2) Doctors with less shame or fear in (1) violating their moral conscience (3) perceive their hospital as more ethically supportive.

Either A or B might be a good title to summarize this article, but the authors’ logic and judgment should explain which one is more correct. 

Readers should be quite interested in this.

This article has some very interesting data and deserves consideration by our journal.

It needs rewriting, however, because many contradictions confuse the reader, viz.:

The title reads “from ethical climate to moral distress,”  and the conclusion says

Line 328:  “Negative affectivity…linking ethical climate to moral distress”

BUT 

Line 361  cautions:  “No causal inferences can be derived”    (TRUE!)

Line 51:  (moral distress) ELICITS frustration, anger, and inadequacy

BUT

Line 78:  powerlessness, frustration, guilt, loss and grief CONTRIBUTE to moral distress.

WHICH direction is the causality here??

Line 58: “The two dimensions (intensity and frequency) have been scarcely studied”

BUT

Line 155:  The Moral Distress Scale-Revised (already used in dozens of studies including notes 21, 49-50-51) itself studies intensity and frequency.

Line 217:  “Participants were mainly physicians  (42.6%) 

This means that the majority were NOT physicians!

Line 43-44: Moral distress means the inability to act morally because of external constraints.

People with moral distress naturally feel ashamed, nervous, afraid, etc.  

The PANAS scale uses emotional categories like ashamed, nervous, afraid, etc.  

Then the question arises:  is this due

(A) to the fact that they feel bad (ashamed or nervous) because they cannot act morally?

(B) Or rather to organizational practices, procedures, and policies (ethical climate)?   

Line 89-90: Ethical climate means perceptions of the ethics-related organizational practices, procedures, and policies. 

BUT

The article regularly refers to “ethical climate” as “positive relationships,” which DIFFER from “perceptions of organizational practices, procedures, and policies.”  This is confusing!

To put it differently, if the “external constraints” in the definition of moral distress equal the “organizational practices, procedures, and policies” in the definition of ethical climate, then ethical climate causes moral distress by definition, regardless of survey questions.

If not, then how does ethical climate differ from “perceptions of organizational practices, procedures, and policies”—without being reduced to “positive relationships”?  Is the meaning of this article that health workers who violate their morality feel less ashamed or nervous if they have supportive colleagues?    Typically, nurses are reported to have more moral distress because they have less authority in decision-making, so shared decision-making can help relieve their moral distress and negative affect.  All the above logic needs clarification and precision.

Sources 74 to 79 should be introduced in the introduction; they form the background to this.

In conclusion, there is much of potential interest in this article, if its definitions are more strictly followed, and its logic more consistently argued.

Reviewer 4 Report

This is a timely study that is well conceived and contributes to this neglected field of inquiry. The definitions of moral distress and ethical relationships are clear, and their relationship are well defined. The conclusions are supported by the data. The conclusions can lead to an action plan to monitor and guide the health care institution culture towards fostering a positive ethical climate;

Reviewer 5 Report

Dear Authors,

I read with pleasure your work. Moral injury among healthcare workers is a topic of great interest since it can deal with poor mental outcomes among healthcare workers. Thus, exploring the variables that can affect moral injury is the base to prevent mental distress among the operators. Your work explores the relationships between moral distress and ethical relationships in the hospital, in a very exposed population such as palliative care workers. 

I find that your manuscript is well written. Introduction explains deeply the constructs you rely on, Methodology is well explained and Discussion makes adequate comparisons between the findings and the existing literature. I only think you should add some comments on the fact that the participants of your study were all volounteer and that can obviously affect the results and their reliabiltiy. Also, as they worked in two different facilities, it should be another limitations due to the fact that the professional enivironment is different. 

Best regards and good luck with your manuscript!

Round 2

Reviewer 1 Report

The comments and concerns of this reviewer, regarding sample size, power of the study,  absence of group discussions, and in-depth interviews were incorporated into the limitations of the study. However, important elements of the study which are vital to the validity and reliability of the study should not be reduced to limitations of the study. The authors should use their own discretion to diversify data visualization, that is, use other methods of data presentation in addition to tables.
